# An Empirical Study of Translation Hypothesis Ensembling with Large Language Models

**António Farinhas**[1,2]  **José G. C. de Souza**[3]  **André F. T. Martins**[1,2,3]

[1]Instituto Superior Técnico (Lisbon ELLIS Unit)

[2]Instituto de Telecomunicações  [3]Unbabel

{antonio.farinhas,andre.t.martins}@tecnico.ulisboa.pt, jose.souza@unbabel.com

## Abstract

Large language models (LLMs) are becoming a one-fits-many solution, but they sometimes hallucinate or produce unreliable output. In this paper, we investigate how hypothesis ensembling can improve the quality of the generated text for the specific problem of LLM-based machine translation. We experiment with several techniques for ensembling hypotheses produced by LLMs such as ChatGPT, LLaMA, and Alpaca. We provide a comprehensive study along multiple dimensions, including the method to generate hypotheses (multiple prompts, temperature-based sampling, and beam search) and the strategy to produce the final translation (instruction-based, quality-based reranking, and minimum Bayes risk (MBR) decoding). Our results show that MBR decoding is a very effective method, that translation quality can be improved using a small number of samples, and that instruction tuning has a strong impact on the relation between the diversity of the hypotheses and the sampling temperature. Our code is available at https://github.com/deep-spin/translation-hypothesis-ensembling.

## 1 Introduction

Significant research effort has been devoted to task-specific neural machine translation (NMT) models trained in a fully supervised manner with large volumes of parallel data. Their performance has been enhanced through techniques such as fine-tuning on in-domain data, model ensembling, and reranking during decoding (Kocmi et al., 2022). The recent achievements of general-purpose large language models (LLMs) such as GPT and LLaMA (OpenAI, 2023; Touvron et al., 2023) offer a fresh perspective on the problem, demonstrating the feasibility of generating high-quality translations without explicit training for the specific task, even in a challenging zero-shot scenario (Hendy et al., 2023).

While techniques such as greedy decoding or sampling from the distribution often prove inadequate for generating translations with task-specific

models, the same cannot be said for LLM-based MT. There is, however, a lack of exploration in this case. Our paper fills this gap by providing a comprehensive study on **ensembling translation hypotheses** (§2), encompassing multiple LLMs such as ChatGPT, LLaMA, and the instruction-tuned Alpaca (Taori et al., 2023). We consider different strategies to generate hypotheses (prompt-based ensembling, temperature sampling, beam search), and several techniques to produce the final translation, including ChooseBest, GenerateBest, reranking based on quality estimation, and minimum Bayes risk decoding. The last two approaches have been successful at improving translation quality with task-specific models (Fernandes et al., 2022; Freitag et al., 2022a), but it is unclear whether the findings hold for LLM-based MT.

Our main findings can be summarized as follows. First, we demonstrate that **translation quality can be enhanced with a small number of samples** (*e.g.*, 20), especially when translating out of English (Fig. 1). In the case of ChatGPT, the cost in terms of paid tokens grows sublinearly with the number of samples (§3.2). Second, we show that similar findings apply to LLaMA and Alpaca (§3.3). We discuss in which conditions beam search remains a reliable baseline for single-hypothesis translation (§3.3.1) and how to ensemble translations (§3.3.2). Moreover, we find that there exists a significant gap in the quality of ensembles of unbiased samples from LLaMA and Alpaca (§3.3.3). We attribute this disparity to how **instruction tuning affects the relationship between the diversity of the hypotheses and the sampling temperature**, which ultimately impacts translation quality. Lastly, we show that **hypothesis ensembling reduces the number of generated hallucinations**, thereby improving the model's robustness to source perturbations (§3.3.4). Ensembling predictions and increasing the model size narrows the quality gap between these models and ChatGPT.

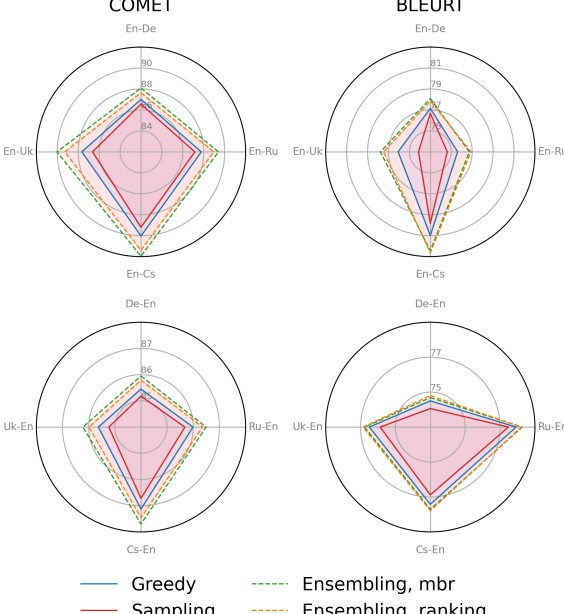

Figure 1: COMET and BLEURT scores for translations produced by ChatGPT. The greedy search output is indicated by a blue bold line, and a single sample baseline by a red bold line. Ensembles of multiple (20) predictions are marked with dashed lines: orange for ranking with COMETKIWI and green for MBR decoding with COMET. Top: EN-X. Bottom: X-EN.

## 2 Ensembling Hypotheses

Ensembling has a long history in machine learning, being well known for leveraging multiple complementary systems to improve performance on a given task and provide good/robust generalization (Hansen and Salamon, 1990; Ting and Witten, 1997; Breiman, 2001; Zhou et al., 2002). While there have been efforts in prompt ensembling and multi-prompt learning within the context of LLMs, this area is largely unexplored for text generation tasks, where the output is a string rather than just a single token (Liu et al., 2023). See §4 for further details. In this section, we delve into the process of generating multiple translation hypotheses (§2.1) and explore different methods for ensembling them to produce a *single* translation (§2.2).[1]

### 2.1 Generating multiple hypotheses

There are several ways of generating multiple predictions from a single language model. In zero-

shot scenarios where no examples are provided in the prompt, we can consider (1) choosing a single prompt and sampling with a temperature such that the resulting predictions are diverse, (2) fixing the sampling temperature and considering multiple prompt templates, or (3) choosing a single prompt that makes the model generate multiple predictions. Refer to App. A.1 for specific prompt templates. While this paper does not cover in-context learning, these strategies can also be applied in few-shot scenarios where in-context examples are provided in the prompt. In such cases, multiple prompts can be created by providing different in-context examples.

### 2.2 Generating a final translation

Let $\bar{\mathcal{Y}} \subseteq \mathcal{Y}$ be a set of $N$ hypotheses, possibly generated with one of the methods discussed in §2.1. When it comes to providing a final output, a commonly used approach involves aggregating the hypotheses in $\bar{\mathcal{Y}}$ by selecting the most frequent one with *majority voting* (Wang et al., 2023a). However, this approach is not well suited for generation tasks such as MT given that the output consists of a sequence of multiple tokens. Therefore, we explore alternative methods that incorporate both external models (§2.2.1) and the LLM itself (§2.2.2).

### 2.2.1 Using external models

Assuming access to an external model $f$ that provides an estimated quality score for a hypothesis $y \in \bar{\mathcal{Y}}$ without requiring a ground truth (*e.g.*, COMETKIWI (Rei et al., 2022b)), a simple approach consists of choosing the hypothesis that maximizes this score (Fernandes et al., 2022),

$$\hat{y}_{\text{ranking}} := \arg\max_{y \in \bar{\mathcal{Y}}} f(y). \qquad (1)$$

Another alternative is minimum Bayes risk (MBR) decoding, which aims to find the output that maximizes an expected *utility* function $u(y^*, y)$ that measures the similarity between a hypothesis $y \in \mathcal{Y}$ and a reference $y^* \in \bar{\mathcal{Y}}$ (Kumar and Byrne, 2002; Eikema and Aziz, 2020). For MT, this can be an automatic evaluation metric such as COMET (Rei et al., 2020). MBR decoding seeks for

$$\hat{y}_{\text{mbr}} := \arg\max_{y \in \bar{\mathcal{Y}}} \mathbb{E}_{Y \sim p_\theta(y|x)}[u(Y, y)], \quad (2)$$

where the expectation in Eq. 2 is typically approximated as a Monte Carlo (MC) sum,

$$\mathbb{E}_{Y \sim p_\theta(y|x)}[u(Y, y)] \approx \frac{1}{M} \sum_{j=1}^{M} u(y^{(j)}, y), \quad (3)$$

---

[1]*Ensembling predictions* in this context should not be confused with the practice of model ensembling, which involves using multiple models (*e.g.*, with different initializations) and combining their outputs. In this paper, we focus on combining hypotheses generated from a single model. The framework remains valid if the hypotheses originate from different models.

using $M$ model samples $y^{(1)}, \ldots, y^{(M)} \sim p_\theta(y|x)$, yielding an *unbiased* MC estimate of the expected utility. Alternatively, we may obtain $y^{(1)}, \ldots, y^{(M)}$ from temperature/nucleus sampling (Holtzman et al., 2020), resulting in a *biased* estimate. While the number of samples used to approximate the expected utility of each hypothesis can be smaller (Eikema and Aziz, 2022), we set $M = N$.

### 2.2.2 Using the LLM

While the techniques above rely on external models for assessing quality, we also propose alternative methods which do not need any external (task-specific) model. We consider two scenarios:[2]

- using the LLM to **select the most appropriate hypothesis** from $\bar{\mathcal{Y}}$ (formulated as a multiple choice question), which we call `ChooseBest`;

- asking the LLM to **generate a final prediction based on the hypotheses** in $\bar{\mathcal{Y}}$ (*i.e.*, a less restrictive scenario where the model has the freedom to either generate a new prediction or to choose one hypothesis from $\bar{\mathcal{Y}}$), which we call `GenerateBest`.

The prompt templates for these methods are provided in App. A.2.

### 2.3 Measuring hypothesis diversity

Inspired by the work of Fomicheva et al. (2020) on quantifying model uncertainty, we measure the *semantic diversity* between different translations for the same source sentence by computing

$$1 - \frac{1}{N(N-1)} \sum_{\substack{i,j=1 \\ j \neq i}}^{N} u(y^{(j)}, y^{(i)}). \qquad (4)$$

It is worth noting that when $u$ is the same utility function used in Eq. 2, this quantity can be computed *without* any additional cost during inference with MBR decoding, as it already provides scores for all the necessary pairwise comparisons.

## 3 Experiments

### 3.1 Setup

We study different methods for generating translations in two regimes:

- A **closed-source** setting using ChatGPT[3], an LLM developed by OpenAI, which has been shown to provide high quality translation (Hendy et al., 2023; Peng et al., 2023a). The system is restricted behind API walls, with undisclosed training data/regime and limited documentation. According to their documentation, it is an InstructGPT model (Ouyang et al., 2022), trained to follow instructions with reinforcement learning from human feedback (Christiano et al., 2017; Stiennon et al., 2020) using proximal policy optimization algorithms (Schulman et al., 2017).[4]

- An **open-source** scenario using LLaMA (Touvron et al., 2023) and Alpaca (Taori et al., 2023). The latter was finetuned from a LLaMA model on an instruction-following dataset with 52K examples generated with `text-davinci-003`, following Wang et al. (2023b). We use the versions with 7B parameters unless otherwise stated.

As our translation baseline, we employ greedy decoding since it generally produces higher-quality outputs, in line with the findings of Peng et al. (2023a), who demonstrate that using a lower sampling temperature leads to improved performance.[5] In this work, we use COMETKIWI (Rei et al., 2022b) for ranking according to Eq. 1 and COMET(-22) (Rei et al., 2022a) as the utility function in MBR decoding, following Eq. 2. We consider 8 different translation directions, including languages such as English (EN), German (DE), Russian (RU), Czech (CS), and Ukrainian (UK). We use the WMT22 test sets (Kocmi et al., 2022), which are recent, and thus less likely to have been part of ChatGPT's training (see footnote 4). Following Freitag et al. (2022b), we evaluate each system with COMET(-22) (Rei et al., 2022a) and BLEURT (Sellam et al., 2020).

### 3.2 Closed-source setting

We generate a set of translation hypotheses for each source sentence by sampling from the model with

---

[2]Another possibility, which we leave for future work, involves a sequential architecture that samples predictions *non-independently* by providing in the prompt the answers generated in previous steps and taking the last prediction as the final output. This is known in statistics as *stacking* (Breiman, 1996) and is related to the work of Madaan et al. (2023).

[3]`https://openai.com/blog/chatgpt`. Our experiments were conducted with the `gpt-3.5-turbo` model between April and June 2023.

[4]For more information, see `https://platform.openai.com/docs/model-index-for-researchers`. According to this, ChatGPT was trained on data from before Q4 2021.

[5]We encountered some API/server errors when prompting ChatGPT for translation with a temperature of 0, as reported by Guerreiro et al. (2023). Using a temperature of 0.1 helps alleviate these issues, which we use as a proxy for greedy decoding. This is not done when using LLaMA/Alpaca.

| N | METHOD | EN-DE | | | EN-RU | | | EN-CS | | | EN-UK | | |
|---|--------|-------|-------|------|-------|-------|------|-------|-------|------|-------|-------|------|
| | | COMET | BLEURT | Cost | COMET | BLEURT | Cost | COMET | BLEURT | Cost | COMET | BLEURT | Cost |
| 1 | Greedy | 87.01 | 77.15 | 1 | 87.77 | 75.61 | 1 | 90.04 | 80.98 | 1 | 87.66 | 76.08 | 1 |
| | Sampling | 86.56 | 76.67 | 1 | 87.15 | 74.62 | 1 | 89.20 | 79.81 | 1 | 86.63 | 74.13 | 1 |
| | *using ChatGPT* | | | | | | | | | | | | |
| 5 | ChooseBest | 87.15 | 77.38 | 6 | 87.77 | 75.47 | 7 | 90.21 | 81.07 | 6 | 87.56 | 75.60 | 7 |
| | GenerateBest | 87.16 | 77.69 | 5 | 87.30 | 75.17 | 6 | 90.05 | 80.89 | 6 | 87.54 | 75.59 | 7 |
| | *using external models* | | | | | | | | | | | | |
| 5 | Ranking | 87.58 | 77.70 | 3 | 88.72 | 76.70 | 3 | 91.02 | 82.14 | 3 | 88.82 | 76.86 | 3 |
| | MBR decoding | 87.77 | 77.71 | 3 | 88.88 | 76.37 | 3 | 91.37 | 81.78 | 3 | 89.23 | 76.87 | 3 |
| | COMET *oracle* | *88.85* | *78.91* | *3* | *89.98* | *77.96* | *3* | *92.18* | *83.07* | *3* | *89.23* | *78.39* | *3* |
| 20 | Ranking | 87.64 | 77.86 | 8 | 88.96 | 76.89 | 10 | 91.40 | 82.64 | 10 | 89.29 | 77.47 | 11 |
| | MBR decoding | 88.09 | 78.09 | 8 | 89.41 | 76.73 | 10 | 91.97 | 82.45 | 10 | 90.03 | 77.77 | 11 |
| | COMET *oracle* | *89.88* | *80.22* | *8* | *90.61* | *79.41* | *10* | *92.26* | *84.55* | *10* | *91.54* | *80.29* | *11* |
| 50 | Ranking | 87.74 | 78.06 | 19 | 89.17 | **77.17** | 24 | 91.52 | **82.80** | 23 | 89.48 | 77.69 | 27 |
| | MBR decoding | **88.25** | **78.14** | 19 | **89.64** | 77.04 | 24 | **92.21** | 82.66 | 23 | **90.31** | **78.03** | 27 |
| | COMET *oracle* | *90.39* | *80.89* | *19* | *91.74* | *80.66* | *24* | *93.75* | *85.29* | *23* | *92.10* | *81.22* | *27* |

| N | METHOD | DE-EN | | | RU-EN | | | CS-EN | | | UK-EN | | |
|---|--------|-------|-------|------|-------|-------|------|-------|-------|------|-------|-------|------|
| | | COMET | BLEURT | Cost | COMET | BLEURT | Cost | COMET | BLEURT | Cost | COMET | BLEURT | Cost |
| 1 | Greedy | 85.45 | 74.50 | 1 | 85.99 | 77.92 | 1 | 87.13 | 77.42 | 1 | 85.63 | 76.50 | 1 |
| | Sampling | 85.18 | 74.06 | 1 | 85.68 | 77.48 | 1 | 86.72 | 76.88 | 1 | 85.23 | 75.88 | 1 |
| | *using ChatGPT* | | | | | | | | | | | | |
| 5 | ChooseBest | 85.37 | 74.43 | 5 | 85.90 | 77.82 | 4 | 87.09 | 77.33 | 4 | 85.51 | 76.30 | 4 |
| | GenerateBest | 85.46 | 74.54 | 4 | 85.66 | 77.56 | 4 | 87.08 | 77.48 | 4 | 85.46 | 76.40 | 4 |
| | *using external models* | | | | | | | | | | | | |
| 5 | Ranking | 85.62 | 74.61 | 2 | 86.22 | 78.08 | 2 | 87.34 | 77.48 | 2 | 85.85 | 76.72 | 2 |
| | MBR decoding | 85.73 | 74.58 | 2 | 86.27 | 78.05 | 2 | 87.49 | 77.51 | 2 | 85.97 | 76.55 | 2 |
| | COMET *oracle* | *86.83* | *75.73* | *2* | *87.46* | *79.57* | *2* | *88.65* | *79.12* | *2* | *87.27* | *78.27* | *2* |
| 20 | Ranking | 85.79 | 74.81 | 6 | 86.36 | 78.22 | 5 | 87.46 | **77.79** | 6 | 85.99 | 76.86 | 5 |
| | MBR decoding | 85.95 | 74.69 | 6 | 86.49 | 78.27 | 5 | 87.70 | 77.71 | 6 | 86.22 | 76.77 | 5 |
| | COMET *oracle* | *87.75* | *76.88* | *6* | *88.42* | *81.03* | *5* | *89.57* | *80.74* | *6* | *88.27* | *79.66* | *5* |
| 50 | Ranking | 85.79 | 74.78 | 13 | 86.32 | 78.13 | 12 | 87.47 | 77.67 | 13 | 85.94 | 76.81 | 11 |
| | MBR decoding | **86.03** | **74.80** | 13 | **86.60** | **78.39** | 12 | **87.79** | 77.73 | 13 | **86.28** | **76.87** | 11 |
| | COMET *oracle* | *88.18* | *77.49* | *13* | *88.95* | *81.86* | *12* | *90.02* | *81.58* | *13* | *88.80* | *80.41* | *11* |

Table 1: Automatic evaluation metrics for ChatGPT and total cost in terms of relative number of tokens normalized within each translation direction, rounded to the nearest unit. Sampling multiple predictions does not increase the prompt's cost, hence the total cost *does not* increase (approximately) linearly with the number of samples. Ranking uses COMETKIWI and MBR decoding uses COMET. Best overall values are **bolded**.

a temperature of 1 (unbiased sampling) using the prompt of Hendy et al. (2023):

> Translate this sentence from [source language] to [target language].
>
> Source: [source sentence].
>
> Target:

We observe that using multiple samples results in at least one significantly higher-quality translation compared to a single prediction, as indicated by automatic evaluation metrics (see oracles for COMET in Table 1). Increasing the number of hypotheses in the set consistently leads to an oracle translation of superior quality, thus highlighting the potential of ensembling predictions.

**Using different prompts.** Based on preliminary experiments conducted on a subset of the language directions mentioned in §3.1, we observed that generating translations using different prompt templates (see App. A.1) yields translations of comparable quality. Furthermore, ensembling predictions from different prompts does not lead to improved results compared to ensembles generated using the same prompt. Therefore, we use only one prompt in our further analysis.

### 3.2.1 How should we ensemble translations?

We compare the methods introduced in §2.2, which include approaches that provide a final answer using only the LLM (ChooseBest and GenerateBest) or that require access to an external model (ranking with COMETKIWI and MBR decoding with COMET as the utility function). Table 1 shows the results for the automatic evaluation with COMET and BLEURT, along with the relative cost of each method compared to greedy decoding. We normalize the values within each translation di-

| N | Method | EN-DE | | EN-RU | | EN-CS | | EN-UK | |
|---|---|---|---|---|---|---|---|---|---|
| | | COMET | BLEURT | COMET | BLEURT | COMET | BLEURT | COMET | BLEURT |
| 1 | LLaMA greedy | 77.33 | 62.86 | 71.57 | 50.80 | 71.56 | 52.33 | 69.06 | 44.47 |
| | Alpaca greedy | 76.67 | 64.06 | 75.59 | 59.52 | 71.40 | 56.61 | 72.76 | 53.40 |
| | LLaMA beam | 77.30 | 59.99 | 62.25 | 34.95 | 71.32 | 45.14 | 61.30 | 29.16 |
| | Alpaca beam | 78.59 | 65.95 | 77.71 | 61.62 | 76.34 | 60.81 | 76.68 | 55.09 |
| | LLaMA unbiased sampling | 51.98 | 36.02 | 42.79 | 21.91 | 42.11 | 21.36 | 42.39 | 20.66 |
| | Alpaca unbiased sampling | 68.15 | 54.86 | 61.50 | 44.15 | 54.90 | 37.45 | 56.80 | 36.44 |
| | LLaMA biased sampling | 69.78 | 55.68 | 63.41 | 42.41 | 60.26 | 39.78 | 59.59 | 36.28 |
| | Alpaca biased sampling | 73.42 | 60.65 | 70.25 | 53.19 | 63.97 | 47.70 | 66.45 | 45.91 |
| | *unbiased sampling* | | | | | | | | |
| 50 | LLaMA ranking | 77.68 | 65.25 | 75.29 | 57.71 | 69.45 | 52.08 | 71.19 | 50.78 |
| | LLaMA MBR decoding | 79.45 | 63.78 | 76.85 | 54.70 | 72.02 | 49.11 | 73.18 | 48.25 |
| | Alpaca ranking | 82.70 | 71.35 | 81.63 | 65.33 | 78.24 | 62.36 | 78.72 | 56.43 |
| | Alpaca MBR decoding | 84.23 | 70.58 | 83.94 | 65.97 | 81.09 | 62.53 | 81.70 | 59.33 |
| | *biased sampling* | | | | | | | | |
| | LLaMA ranking | 83.04 | 71.91 | 82.93 | 67.41 | 81.07 | 66.88 | 81.12 | 62.24 |
| | LLaMA MBR decoding | 84.06 | 69.84 | 83.72 | 64.75 | 82.87 | 63.61 | 82.01 | 60.14 |
| | Alpaca ranking | 83.58 | **72.30** | 84.12 | **68.75** | 82.42 | **68.50** | 82.22 | 61.09 |
| | Alpaca MBR decoding | **84.54** | 71.18 | **85.44** | 68.32 | **84.82** | 68.16 | **84.30** | **63.42** |

| N | Method | DE-EN | | RU-EN | | CS-EN | | UK-EN | |
|---|---|---|---|---|---|---|---|---|---|
| | | COMET | BLEURT | COMET | BLEURT | COMET | BLEURT | COMET | BLEURT |
| 1 | LLaMA greedy | 82.36 | 70.19 | 81.58 | 71.62 | 81.26 | 69.90 | 81.37 | 71.13 |
| | Alpaca greedy | 82.31 | 70.14 | 81.65 | 71.89 | 81.14 | 69.69 | 81.34 | 70.90 |
| | LLaMA beam | 82.56 | 70.49 | 82.19 | 72.50 | 82.08 | 70.83 | 81.97 | 71.99 |
| | Alpaca beam | 82.53 | 70.40 | 82.08 | 72.29 | 81.69 | 70.26 | 81.55 | 71.09 |
| | LLaMA unbiased sampling | 73.26 | 60.23 | 70.63 | 58.62 | 70.19 | 57.20 | 70.72 | 59.13 |
| | Alpaca unbiased sampling | 81.04 | 68.66 | 79.92 | 69.62 | 79.03 | 67.20 | 79.52 | 68.85 |
| | LLaMA biased sampling | 79.82 | 67.60 | 78.75 | 68.14 | 78.10 | 66.07 | 78.71 | 67.84 |
| | Alpaca biased sampling | 81.86 | 69.71 | 81.09 | 71.02 | 80.44 | 68.81 | 80.72 | 70.33 |
| | *unbiased sampling* | | | | | | | | |
| 50 | LLaMA ranking | 82.90 | 70.64 | 82.12 | 71.74 | 81.85 | 69.97 | 82.17 | 71.70 |
| | LLaMA MBR decoding | 84.25 | 70.75 | 83.22 | 71.61 | 82.92 | 69.62 | 83.40 | 71.51 |
| | Alpaca ranking | 83.97 | 72.22 | 83.62 | 74.07 | 83.75 | 72.79 | 83.40 | 73.47 |
| | Alpaca MBR decoding | 84.47 | 71.78 | 83.95 | 73.50 | 84.00 | 71.78 | 83.58 | 72.46 |
| | *biased sampling* | | | | | | | | |
| | LLaMA ranking | 84.03 | 72.15 | 83.44 | 73.73 | 83.58 | 72.32 | 83.50 | 73.47 |
| | LLaMA MBR decoding | **85.03** | 72.17 | **84.22** | 73.30 | **84.4** | 71.87 | **84.23** | 72.82 |
| | Alpaca ranking | 84.10 | **72.43** | 83.70 | **74.32** | 83.92 | **72.97** | 83.56 | **73.67** |
| | Alpaca MBR decoding | 84.02 | 71.31 | 83.56 | 73.33 | 83.61 | 71.47 | 83.08 | 72.22 |

Table 2: Automatic evaluation metrics for LLaMA (7B) and Alpaca (7B). Ranking uses COMETKIWI and MBR decoding uses COMET. Best overall values are **bolded** and best within each group are underlined.

rection by dividing by the cost of greedy decoding.

**Using ChatGPT.** Although the performance of `ChooseBest` and `GenerateBest` with 5 samples is slightly better than the single sample baseline, these approaches still fall short of both the greedy decoding output and the methods that use external models for selecting the final translation, according to both COMET and BLEURT. Furthermore, the incorporation of all translation hypotheses in the prompt (see App. A.2) significantly increases the cost, making these approaches less scalable. For that reason, we chose not to pursue this direction further and instead focused our efforts on exploring the methods described in §2.2.1.

**Using external models.** Table 1 shows that the two methods that use external models for ensembling predictions are effective at increasing the final translation quality over the baselines. Notably, these methods achieve significant improvements without requiring an extensive number of *unbiased* samples from the model's distribution, especially when translating out of English. Fig. 1 provides a visual representation of the gains achievable with 20 samples. This differs from the findings of previous research using task-specific NMT models (Fernandes et al., 2022; Freitag et al., 2022a), where it is typically necessary to bias the model's distribution using techniques like nucleus sampling (Holtzman et al., 2020) or to train models without

label smoothing ([Eikema and Aziz, 2020](); [Freitag et al., 2022a]()), which often leads to an impractical increase in cost due to the need for more translation hypotheses. MBR decoding consistently achieves the best results according to COMET across all translation directions. Although the differences in quality are small, this pattern does not hold when evaluating with BLEURT for EN-RU and EN-CS, for which ranking with COMETKIWI appears to have an edge.

### 3.3 Open-source setting

We obtain sets of both *biased* and *unbiased* translation hypotheses for each source sentence from LLaMA and Alpaca. The former is obtained by sampling with a temperature of 1, while the latter uses temperature and nucleus sampling (with $t = 0.8$ and $p = 0.95$), which are the defaults for LLaMA.[6] We use a variation of the prompt of [Hendy et al. (2023)]() which stresses the translation direction (crucial for the non instruction-tuned LLaMA to understand the task) as follows,

> Translate this sentence from [source language] to [target language].
>
> [source language] Source: [source sentence].
>
> [target language] Translation:

While most works on translation with general-purpose LLMs typically present results using unbiased sampling or **greedy decoding**, as it has been observed that reducing the sampling temperature generally enhances translation quality ([Peng et al., 2023a]()), it is worth exploring the impact of using **beam search** ([Reddy, 1977]()), the go-to search strategy for decoding with task-specific models. Thus, in addition to greedy decoding, we employ beam search (with a beam size of 5) as a single hypothesis baseline. Notably, this is not possible for ChatGPT (§3.2) because its API does not include beam search. We report results in Table 2, and use them to answer specific research questions next.

#### 3.3.1 Greedy vs. beam search baseline

Fig. 2 compares greedy search and beam search for both LLaMA and Alpaca for X-EN (right) and EN-X (left) translation tasks. For X-EN, beam search outperforms greedy search, with LLaMA achieving the highest overall quality. However, the

[6]We use the implementation in https://github.com/facebookresearch/llama/tree/llama_v1.

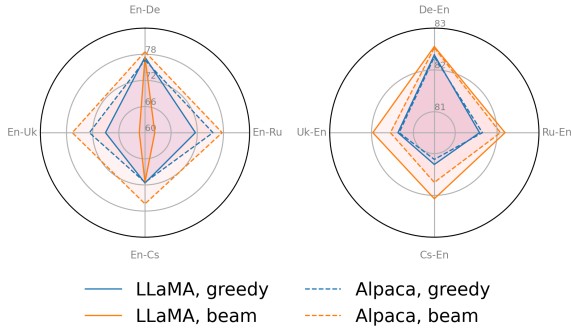

Figure 2: COMET scores for LLaMA and Alpaca with greedy (blue) and beam search (orange). We represent LLaMA with solid lines and Alpaca with dashed lines. Left: EN-X. Right: X-EN.

|  |  | EN-DE | EN-RU | EN-CS | EN-UK |
|---|---|---|---|---|---|
| LLaMA | Greedy | 11.2 | 25.0 | 21.5 | 31.6 |
|  | Beam | 25.2 | 43.4 | 46.4 | 41.2 |
|  | Ranking | 1.3 | 2.5 | 3.6 | 11.3 |
|  | MBR | 6.4 | 6.4 | 8.8 | 12.1 |
| Alpaca | Greedy | 2.1 | 2.7 | 4.5 | 13.6 |
|  | Beam | 4.0 | 6.4 | 10.6 | 21.3 |
|  | Ranking | **0.4** | 1.8 | 3.1 | 17.1 |
|  | MBR | 1.4 | **0.8** | **2.6** | **10.7** |

Table 3: Percentage of translations in the wrong target language when translating from EN. Ranking with COMETKIWI and MBR decoding with COMET use biased samples. We do not show the values for X-EN given that only a few translations (<1% for all languages) are not in EN. Best overall values are **bolded** and best for each model are underlined.

results are not as favorable when applying beam search to the non instruction-tuned LLaMA for EN-X (particularly for EN-RU and EN-UK). In contrast, for Alpaca, beam search is consistently better in both language directions.

We hypothesize that this discrepancy is related to LLaMA's relative inability to generate text in languages other than English. To validate this hypothesis, we automatically identify the language of the provided translations using the language identification model of [Joulin et al. (2016a,b)](). While both LLaMA and Alpaca, with both greedy and beam search, correctly provide translations in English when requested (X-EN), the same cannot be said for other languages (see Table 3). In particular, LLaMA frequently provides translations in the wrong target language (mostly in English), especially with beam search. Interestingly, ensembling predictions (*e.g.*, with ranking or MBR decoding) effectively mitigates this issue. Providing a few in-context examples, which we leave for future

work, is another alternative that may help improve LLaMA's performance, alleviating the impact of decoding with alternative methods.

### 3.3.2 How should we ensemble translations?

Table 2 shows that, for all models and translation directions, a single sample baseline is not competitive *at all* with the greedy and beam search outputs, with the latter achieving the best overall quality, as discussed in §3.3.1. Sampling's poor performance is, however, more noticeable when translating from English. Given that the overall quality scores are lower than that of ChatGPT (Table 1), in App. B we report results for a larger version of LLaMA (with 30B parameters), for which we also observe performance gains from ensembling translations. Besides, App. C contains additional results considering a few-shot learning scenario where in-context examples are provided in the prompt.

For EN-X, ensembles of unbiased samples from LLaMA do not perform well, a topic we will further study in §3.3.3. Overall, Alpaca performs better, and the final quality of the ensemble can be boosted by biasing the samples (although the difference is not very significant for EN-DE). MBR decoding with COMET attains the best results in terms of COMET, while ranking with COMETKIWI is better in terms of BLEURT for most language pairs.

For X-EN, while biased sampling is still advantageous and the best results in terms of BLEURT are still obtained with Alpaca (ranking with COMETKIWI), the best COMET scores are attained using LLaMA (MBR decoding).

### 3.3.3 Biasedness, diversity, and quality

There exists a significant gap in the final quality of an ensemble of unbiased samples from LLaMA and Alpaca, especially in the case of EN-X translations, where LLaMA's performance is notably poor. For example, as shown in Table 2, the disparity in COMET and BLEURT scores for EN-DE is 5 and 7 points, respectively. In this section, we study how **instruction tuning** influences the relationship between candidate diversity and sampling temperature, and its impact on final translation quality. We consider translations from English to German (see App. D for the reversed direction) as a case study and measure translation diversity using the method described in §2.3, with COMET as the similarity function $u$ in Eq. 4.

Fig. 3 shows how the final translation quality, represented by the green and orange lines, and

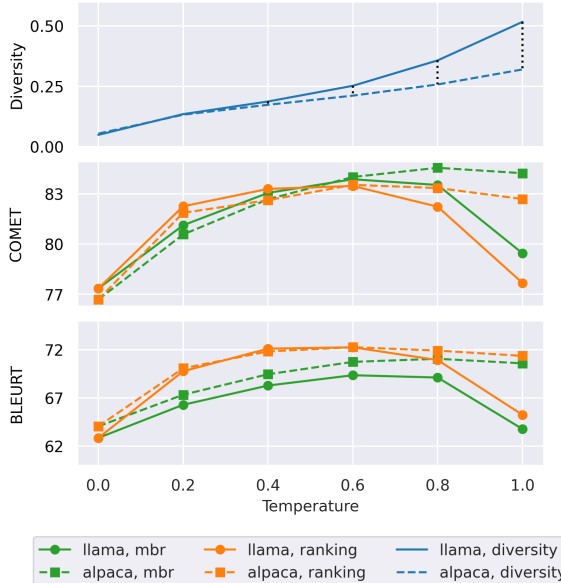

Figure 3: Values for BLEURT (bottom) and COMET (middle) for MBR decoding with COMET (green) and ranking with COMETKIWI (orange), and diversity between hypotheses (top; blue) as we increase the sampling temperature for EN-DE. We represent LLaMA with solid lines and Alpaca with dashed lines. The dotted black lines (top) mark the increasing diversity gap.

the diversity between hypotheses, depicted by the blue lines, vary with the sampling temperature for LLaMA (solid lines) and for Alpaca (dashed lines). As expected, the diversity between hypotheses increases as the sampling temperature increases. However, this occurs at a *different rate* for LLaMA and Alpaca, indicating that instruction tuning changes the relationship between hypothesis diversity and sampling temperature. Ultimately, this affects the final quality of the ensemble, which may help explain the aforementioned quality gap for ensembles of unbiased samples.

An interesting observation is the noticeable increase in the diversity gap (*i.e.*, the length of the dotted black lines increases for temperatures ranging from 0.6 to 1.0 in Fig. 3), which coincides with a divergence in the translation quality between LLaMA and Alpaca ensembles (the solid and dashed lines begin to separate). Additionally, it is worth noting that the optimal COMET scores are attained at a candidate diversity of approximately 0.25 for both models; however, this optimum corresponds to different sampling temperatures for each model.

Overall, we conclude that instruction tuning has a notable impact on the relation between hypotheses diversity and sampling temperature, influencing

|        |              | EN-CS | EN-UK |
|--------|--------------|-------|-------|
| LLaMA  | Greedy       | 10.2  | 21.2  |
|        | Ranking      | 2.3 | 5.2 |
|        | MBR decoding | 6.9   | 7.4   |
| Alpaca | Greedy       | 2.3   | 5.8   |
|        | Ranking      | **0.8** | 4.1   |
|        | MBR decoding | 1.2   | **2.2** |

Table 4: Rate of hallucinations (the percentage is over the number of sentences that passed the quality threshold for non-perturbed sources). Ranking with COMETKIWI and MBR decoding with COMET use biased samples. Best overall values are **bolded** and best for each model are underlined.

the final translation quality. Notably, it is simpler to set an appropriate temperature for the instruction-tuned Alpaca, as it is less sensitive to such variations. We observe that this effect is less pronounced when translating into English (refer to App. D), likely due to the higher inherent similarity between hypotheses—potentially attributable to the extensive English training data used for these models.

### 3.3.4 Hallucinations under perturbation

In this section, we study how robust LLaMA and Alpaca are to perturbations in the source text by searching for hallucinations under perturbation, which correspond to situations where a model produces drastically different translations for unperturbed and slightly perturbed inputs (Lee et al., 2019; Raunak et al., 2021). We focus on EN-CS and EN-UK translations, given that hallucinations are typically more frequent when translating out of English and for lower resource languages. We follow Guerreiro et al. (2023) and apply the minimal perturbations of Xu et al. (2023), including misspelling and title-casing words, and inserting frequent tokens at the beginning of the source sentence. See App. E for further details.

Table 4 shows that the hallucination rates decrease with instruction tuning for both EN-CS and EN-UK. Ensembling translation hypotheses further decreases the number of hallucinations, suggesting that considering multiple hypotheses is a promising method for alleviating this issue.

## 4 Related Work

**Ensembling.** Recently, Peng et al. (2023b) compare the effectiveness of different model ensemble strategies but focus on trained *soft* prompts and do not explore generation tasks. There is also work on ensembling predictions (produced by either sam-

pling multiple times or by using different prompts) with majority voting (Wang et al., 2023a; Liévin et al., 2023; Diao et al., 2023), which is not really suited for MT as argued before, or along with other complementary approaches (Wang et al., 2022; Li et al., 2022; Sun et al., 2023). There are several works on ensembling for NMT, where a decoder uses multiple models (*e.g.*, with different initializations) and predicts an output by averaging token-level predictions from each model (Sutskever et al., 2014; Chung et al., 2016), whereas our approach considers full translations from a single model.

**Ranking/rescoring hypotheses.** Garcia et al. (2023) train their own language models, sample multiple hypotheses and choose a final translation using MBR decoding, which has been shown to improve the translation capabilities of task-specific models (Fernandes et al., 2022; Freitag et al., 2022a). Their work is significantly different from ours, since their models exclusively support two or three languages at a time. Similarly, the approach of Yang et al. (2022) includes a reranking stage (including two trained dedicated rerankers) and an edit stage, while Kadavath et al. (2022) ask models to directly evaluate the probability that their self-generated answers are correct.

**Editing/refining hypotheses.** Raunak et al. (2023) explore translation editing with LLMs but they do not study how to use external models (*e.g.* COMET and COMETKIWI) to improve translation quality. Similarly, Chen et al. (2023) propose to iteratively refine translations using LLMs, attaining comparable translation quality with the baseline, according to automatic translation metrics, and reducing *translationese*, according to a human study.

## 5 Conclusions and Future Work

We have conducted a thorough empirical analysis on various techniques for generating translations using LLMs. Our study encompasses eight datasets and three model classes, including closed-source and open-source models, the latter with and without instruction tuning. We have demonstrated that ensembling predictions significantly enhances translation quality and reduces hallucinations under source perturbations. Additionally, we have discovered that instruction tuning affects the relationship between the diversity of sampled hypotheses and the sampling temperature, which in turn influences the final translation quality.

There are several avenues for future research in addition to the ones that have already been mentioned in previous sections. While ensembling predictions produced by LLMs is effective at improving translation quality, it also presents opportunities for developing improved methods of uncertainty quantification and calibration (Fomicheva et al., 2020; Tian et al., 2023), crucial for addressing the inherent opacity of black-box LLMs.

## Limitations

We highlight three main limitations of our work. First, we primarily focus on versions of LLaMA and Alpaca with 7B parameters, even though we have included additional results using a model with 30B parameters in App. B. It remains unclear how our findings, such as the effects of employing beam search (§3.3.1), or how instruction tuning influences the relationship between sampling temperature and hypothesis diversity (§3.3.3), generalize to *even larger* models.

Second, we have included results from ChatGPT due to its proven ability to provide high-quality translation. Notably, ChatGPT is a restricted system accessible only through APIs, and its training data/regime are undisclosed. Since there is limited documentation, it is difficult to ensure that ChatGPT did not encounter our evaluation benchmarks during training, even though they are recent (§3.1).

Lastly, due to the high cost and time required for conducting a final human assessment of the translation quality, we have not included it in our evaluation. Instead, we try to address this issue by reporting results based on multiple state-of-the-art automatic evaluation metrics for machine translation, such as COMET and BLEURT. Despite these limitations, we believe that our findings hold significance for the ML/NLP community.

## Ethics Statement

ChatGPT and Alpaca have been finetuned using instructions and/or human feedback, which may be low-quality, contradictory, or adversarial, possibly resulting in inherent biases (Fernandes et al., 2023). For example, instructions may lack specificity, leading annotators to inadvertently evaluate a slightly different task (Parmar et al., 2023). Another concern arises from using quality estimation/evaluation models such as COMETKIWI and COMET, which have been finetuned on human preferences. In such cases, annotators may fail to consider better alternatives when presented with a given text, resulting in the misclassification of isolated text as high quality (Bansal et al., 2021). Additionally, all evaluation benchmarks used in this study are openly accessible, and annotators were allowed to label sensitive information when necessary. Lastly, it is important to note that all LLMs exhibit a shared concern regarding their energy consumption, particularly during the training phase (Strubell et al., 2019).

## Acknowledgments

We would like to thank Pedro Martins, Nuno Guerreiro, Ben Peters, John Mendonça, Duarte Alves, Sweta Agrawal, and the SARDINE lab team for helpful discussions. This work was built on open-source software; we acknowledge Van Rossum and Drake (2009); Oliphant (2006); Walt et al. (2011), and (Paszke et al., 2019). This work was supported by EU's Horizon Europe Research and Innovation Actions (UTTER, contract 101070631), by the project DECOLLAGE (ERC-2022-CoG 101088763), by the Portuguese Recovery and Resilience Plan through project C645008882- 00000055 (Center for Responsible AI), and by Fundação para a Ciência e Tecnologia through contract UIDB/50008/2020.

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

## A  Prompt Templates

### A.1  Translation

In addition to the prompt of Hendy et al. (2023), used in the reported experiments with ChatGPT,

> Translate this sentence from [`source language`] to [`target language`].
>
> Source: [`source sentence`].
>
> Target:

we also tried the prompt of Peng et al. (2023a),

> Please provide the [`target language`] translation for this sentence: [`source sentence`]

the prompt of Gao et al. (2023), which provides additional information on the translation task and the language pairs involved,

> This is a [`source language`] to [`target language`] translation task, please provide the [`target language`] translation for this sentence: [`source sentence`]

and the prompt of Zhang et al. (2023), which is simpler but concise,

> [`source language`]:      [`source sentence`]
>
> [`target language`]:

In a preliminary stage of this work, we observed that the results according to automatic evaluation metrics were similar for all the prompts above. In addition, ensembling translations generated with multiple prompts was not better than sampling hypotheses using a single prompt template. We also attempted to generate multiple translations (N) with the following prompt,

> Translate this sentence from [`source language`] to [`target language`] in [`N`] different ways.
>
> Source: [`source sentence`]
>
> [`N`] translations:

However, we observed a decline in translation quality as the model generated subsequent translations, with the first one exhibiting lower quality compared to the ones generated using the prompts mentioned above. For example, in the case of EN-DE translation, there was an approximate gap of 3 COMET points between the first and last translation. For that reason, we decided to discard this approach.

For LLaMA and Alpaca, we use a variation of the prompt of Hendy et al. (2023) which stresses the translation direction (crucial for the non instruction-tuned LLaMA to understand the task) as follows,

> Translate this sentence from [`source language`] to [`target language`].
>
> [`source language`] Source: [`source sentence`].
>
> [`target language`] Translation:

### A.2  Generation of the final translation

We formulate the task of choosing the most appropriate hypothesis (`ChooseBest`) as a multiple choice question using the following prompt,

> This is a multiple choice question, choose a single answer. What is the best [`target language`] translation for this [`source language`] sentence?
>
> Source: [`source`]
>
> Option A. [`hypothesis 1`]
>
> Option B. [`hypothesis 2`]
>
> ...
>
> Correct answer: Option

Besides, we ask the LLM to generate a final prediction based on the provided hypotheses (`GenerateBest`) using the following prompt,

> Use the following translation hypotheses to generate the best possible [`target language`] translation for this [`source language`] sentence.
>
> Source: [`source`]
>
> Translation hypotheses:
>
> [`hypothesis 1`]
>
> [`hypothesis 2`]
>
> ...
>
> Best possible translation:

| N | METHOD | EN-DE | | EN-RU | | EN-CS | | EN-UK | |
|---|--------|-------|------|-------|------|-------|------|-------|------|
| | | COMET | BLEURT | COMET | BLEURT | COMET | BLEURT | COMET | BLEURT |
| 1 | ChatGPT greedy | 87.01 | 77.15 | 87.77 | 75.61 | 90.04 | 80.98 | 87.66 | 76.08 |
| 1 | Greedy | _81.54_ | _69.39_ | _82.35_ | _67.27_ | _82.17_ | _69.27_ | _81.32_ | _66.00_ |
| | Biased sampling | 77.47 | 64.87 | 75.89 | 58.62 | 73.59 | 58.47 | 74.29 | 56.50 |
| 20 | Ranking | 84.87 | _73.91_ | 86.02 | _72.11_ | 87.02 | _75.48_ | 85.64 | _71.36_ |
| | MBR decoding | _85.78_ | 73.26 | _87.05_ | 71.45 | _88.45_ | 74.71 | _86.68_ | 70.61 |
| | COMET *oracle* | *87.74* | *75.83* | *88.85* | *74.13* | *89.70* | *76.39* | *88.51* | *73.41* |
| 50 | Ranking | 85.16 | **74.37** | 86.75 | **72.93** | 88.11 | **76.84** | 86.29 | **71.61** |
| | MBR decoding | **86.48** | 73.97 | **87.79** | 72.37 | **89.61** | 76.33 | **87.62** | 71.52 |
| | COMET *oracle* | *88.77* | *77.12* | *90.02* | *76.09* | *91.10* | *78.76* | *89.80* | *75.60* |
| | | DE-EN | | RU-EN | | CS-EN | | UK-EN | |
| | | COMET | BLEURT | COMET | BLEURT | COMET | BLEURT | COMET | BLEURT |
| 1 | ChatGPT greedy | 85.45 | 74.50 | 85.99 | 77.92 | 87.13 | 77.42 | 85.63 | 76.50 |
| 1 | Greedy | _83.12_ | _71.01_ | _83.00_ | _73.48_ | _83.64_ | _72.60_ | _82.86_ | _72.50_ |
| | Biased sampling | 81.29 | 69.03 | 80.83 | 70.56 | 81.23 | 69.50 | 80.67 | 69.89 |
| 20 | Ranking | 84.68 | _72.85_ | 84.43 | 74.83 | 85.24 | _74.45_ | 84.69 | _74.36_ |
| | MBR decoding | _85.18_ | 72.64 | _85.06_ | _75.00_ | _85.74_ | 73.91 | _84.91_ | 73.87 |
| | COMET *oracle* | *87.33* | *75.70* | *87.22* | *78.14* | *87.89* | *77.13* | *87.62* | *77.78* |
| 50 | Ranking | 84.76 | 72.96 | 84.53 | 75.05 | 85.45 | **74.46** | 84.70 | **74.77** |
| | MBR decoding | **85.48** | **72.97** | **85.34** | **75.18** | **86.17** | 74.29 | **85.33** | 74.29 |
| | COMET *oracle* | *88.13* | *76.93* | *88.08* | *79.45* | *88.76* | *78.60* | *88.65* | *79.36* |

Table 5: Automatic evaluation metrics for LLaMA (30B). We use temperature and nucleus sampling (with $t = 0.8$ and $p = 0.95$), which is the default (see §3.3). Ranking uses COMETKIWI and MBR decoding uses COMET. Best overall values for LLaMA are **bolded** and best within each group are underlined. Values for ChatGPT with greedy decoding are taken from Table 1 and highlighted in red.

| FEW-SHOT | METHOD | EN-DE | | EN-RU | |
|----------|--------|-------|------|-------|------|
| | | COMET | BLEURT | COMET | BLEURT |
| 0 | Greedy | 77.33 | 62.86 | 71.57 | 50.80 |
| | Ranking | 83.04 | _71.91_ | 82.93 | _67.41_ |
| | MBR decoding | _84.06_ | 69.84 | _83.72_ | 64.75 |
| | COMET *oracle* | *86.51* | *73.42* | *86.59* | *69.43* |
| 5 | Greedy | 79.82 | 68.02 | 80.20 | 65.01 |
| | Ranking | 83.72 | **72.90** | 84.98 | **70.38** |
| | MBR decoding | **85.42** | 72.66 | **86.64** | 70.37 |
| | COMET *oracle* | *87.42* | *75.06* | *88.43* | *72.88* |

Table 6: Automatic evaluation metrics for LLaMA (7B) with and without few-shot learning. For ranking with COMETKIWI and MBR decoding with COMET we use 50 samples obtained through temperature and nucleus sampling (with $t = 0.8$ and $p = 0.95$), which is the default (see §3.3). Best overall values are **bolded** and best within each group are underlined.

## B    Increasing Model Size

Table 5 shows COMET and BLEURT scores for LLaMA (30B), along with the ChatGPT output (§3.2). The main findings discussed in §3.2 still hold: ensembling multiple translations is effective at improving the overall translation quality. Notably, these quality scores are more competitive with the ChatGPT baseline, suggesting that increasing the number of model's parameters is beneficial even without instruction tuning.

## C    Few Shot Learning

As discussed in §2.1, the candidate generation strategies described in our paper can also be applied in few-shot scenarios where in-context examples are provided in the prompt. While this paper

does not focus on this case and covers other orthogonal dimensions in more detail (*e.g.*, choice of model, method to generate hypotheses, strategy to generate the final translation), we include results with few-shot examples on a subset of the language pairs (EN-DE and EN-RU), in Table 6. We consider 5-shot examples from the FLORES-200 dev set (Guzmán et al., 2019; Goyal et al., 2022; Team et al., 2022) and use LLaMA (7B). As expected, we see that hypothesis ensembling still works well for such a setting, with the overall scores being higher than in a 0-shot scenario.

## D  Biasedness, Diversity, and Quality

Fig. 4 shows that the trends observed in Fig. 3 for EN-DE translations are also observable in the reversed translation direction (DE-EN). Once again, *increasing* the sampling temperature leads to an *increase* in the diversity of the hypotheses, and this trend varies at different rates for LLaMA and Alpaca. As the diversity gap increases, the translation quality between ensembles of samples from these models diverges. However, it is worth noting that this effect is less pronounced for DE-EN, likely due to the extensive English training data used for these models that results in lower absolute values of translation diversity, as indicated by the blue lines.

## E  Hallucinations

We follow the choices of Guerreiro et al. (2023) and detect hallucinations under perturbations as follows. For each language pair, we start by obtaining source sentences for which all methods (greedy, ranking with COMETKIWI, and MBR decoding with COMET) generate unperturbed translations that meet a minimum quality threshold (BLEU $> 9$). Then, we set a low maximum quality score for perturbed translations (BLEU $< 3$). A model generates a hallucination when both thresholds are met. The metric for measuring lexical overlap and the threshold values we used follow previous work.

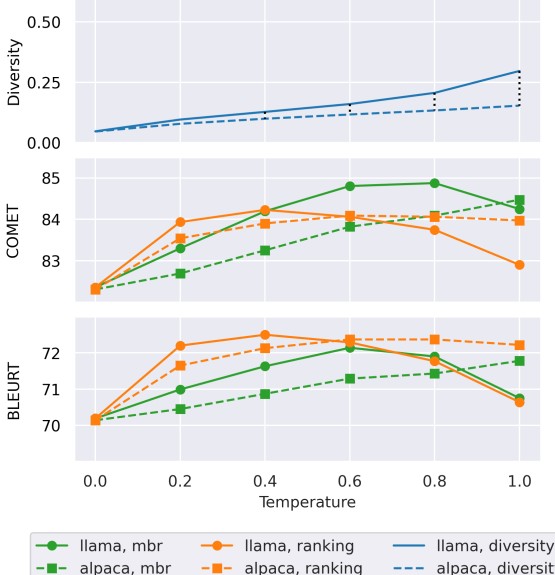

Figure 4: Values for BLEURT (bottom) and COMET (middle) for MBR decoding with COMET (green) and ranking with COMETKIWI (orange), and diversity between hypotheses (top; blue) as we increase the sampling temperature for DE-EN. We represent LLaMA with solid lines and Alpaca with dashed lines. The dotted black lines (top) mark the increasing diversity gap.