# OpenReview forum: "An Empirical Study of Translation Hypothesis Ensembling with Large Language Models"
_EMNLP/2023/Conference — EMNLP 2023 Main_

### Official Review · Reviewer_UV4z · 2023-08-04

**Soundness:** 3

**Excitement:**

3: Ambivalent: It has merits (e.g., it reports state-of-the-art results, the idea is nice), but there are key weaknesses (e.g., it describes incremental work), and it can significantly benefit from another round of revision. However, I won't object to accepting it if my co-reviewers champion it.

**Paper Topic And Main Contributions:**

This paper studies how to improve LLM-based MT by reranking multiple hypotheses. In the experiment, MBR decoding with COMET improved translation performance.

This study shows that the simple approach can boost the performance of LLM-based MT through the experiments for various language pairs, and the result may be useful. On the other hand, the approach tried in the paper is not novel, and it is not clear if it works for other settings.


**Questions For The Authors:**

In the experiment, the LLM based approaches (ChooseBest and GenerateBest) did not perform well. This looked somewhat surprising considering that a good result about LLM-based quality estimation is recently reported (Kocmi and Federmann, Large Language Models Are State-of-the-Art Evaluators of Translation Quality). It will be better to discuss why the approaches did not work.


**Reasons To Accept:**

- The paper shows that the performance of LLM-based MT is improved with simple methods.
- The experiments are conducted for 8 language pairs using multiple LLMs.


**Reasons To Reject:**

- The approach is not novel. MBR decoding is widely known. The paper tries two simple approaches using LLMs but they did not work well.
- LLMs are often used with few-shot prompts, and it is not clear if the hypothesis reranking works well for such a setting.


**Reproducibility:**

4: Could mostly reproduce the results, but there may be some variation because of sample variance or minor variations in their interpretation of the protocol or method.

**Reviewer Confidence:**

3: Pretty sure, but there's a chance I missed something. Although I have a good feel for this area in general, I did not carefully check the paper's details, e.g., the math, experimental design, or novelty.

**Typos Grammar Style And Presentation Improvements:**

I felt the term "ensembling" in this paper is confusing. In the context of ML, ensembling means to combine multiple models. This paper considers to select the best hypothesis from multiple candidates produced by a single model, which is often called reranking.

---

> ### Author Rebuttal · Authors · 2023-08-28
>
> Thank you for the review and suggestions. We address below your main concerns about our paper.
>
> > The approach tried in the paper is not novel, and it is not clear if it works for other settings.
>
> Please note that while MBR decoding and ranking approaches based on neural quality estimation metrics have been studied before for task-specific NMT models (a line of research which usually considers large sets of hypotheses), our paper focuses on a understudied and increasingly relevant scenario, LLM-based translation using general-purpose language models and a relatively small number of hypotheses. Our study provides  important  insights about  what changes in this scenario. For instance, contrary to previous findings using task-specific models, our experiments suggest that even a small number of samples can effectively improve performance; we also discuss in which conditions we do not need biased samples (see $\S 3.2.1$ and $\S 3.3.3$), streamlining the method's applicability. In addition, we believe that our paper sheds light on how to generate high-quality translations with LLMs, including in which conditions beam search is a reliable baseline ($\S 3.3.1$). More notably, we find that hypothesis ensembling reduces the number of generated hallucinations (see $\S 3.3.4$) and thus improves the model's robustness to source perturbations.
>
> > LLMs are often used with few-shot prompts, and it is not clear if the hypothesis reranking works well for such a setting.
>
> As discussed in $\S 2.1$, the candidate generation strategies described in our paper can also be applied in few-shot scenarios where in-context examples are provided in the prompt. We chose not to focus much on this scenario  so that we could  cover  other (orthogonal) dimensions in more detail  (e.g., choice of model, method to generate hypotheses, strategy to generate the final translation). However, we agree that including a small experiment with few-shot examples can be valuable, and therefore we decided to run some experiments to answer your question. As expected, we found that hypothesis ensembling still works well for such a setting (the numbers below are for 5-shot examples from the FLORES-200 dev set using LLaMA 7B (En-De and En-Ru)):
>
> |                  |         |    | En-De     |           | En-Ru     |           |
> |------------------|---------|----|-----------|-----------|-----------|-----------|
> | Model            | Method  | N  | COMET     | BLEURT    | COMET     | BLEURT    |
> | LLaMA 7B, 0-shot | Greedy  | 1  | 77.33     | 62.86     | 71.57     | 50.80     |
> | LLaMA 7B, 0-shot | Ranking | 50 | 83.04     | 71.91     | 82.93     | 67.41     |
> | LLaMA 7B, 0-shot | MBR     | 50 | 84.06     | 69.84     | 83.72     | 64.75     |
> | LLaMA 7B, 5-shot | Greedy  | 1  | 79.82     | 68.02     | 80.20     | 65.01     |
> | LLaMA 7B, 5-shot | Ranking | 50 | 83.72     | **72.90** | 84.98     | **70.38** |
> | LLaMA 7B, 5-shot | MBR     | 50 | **85.42** | 72.66     | **86.64** | 70.37     |
>
> We will update our paper accordingly.
>
> > In the experiment, the LLM based approaches (ChooseBest and GenerateBest) did not perform well. This looked somewhat surprising considering that a good result about LLM-based quality estimation is recently reported (Kocmi and Federmann, Large Language Models Are State-of-the-Art Evaluators of Translation Quality). It will be better to discuss why the approaches did not work.
>
> This is an interesting point. While Kocmi and Federmann report state-of-the-art performance on assessing system-level quality (see their Table 2), the same cannot be said for segment-level evaluation with ChatGPT (see their Table 4), in which GPT-based methods still lag behind dedicated evaluation systems like COMET or BLEURT. Notably, methods like GenerateBest (proposed in our paper) depend on the capabilities of systems to perform reliable segment-level assessments, which may explain the poor performance of these methods. However, it is important to note that machine translation evaluation/quality estimation using LLMs remains an area of active research, and we expect that the landscape might change as more advanced models/methods (with improved segment-level capabilities) are introduced. We will add this discussion to the paper.
>
> > I felt the term "ensembling" in this paper is confusing. In the context of ML, ensembling means to combine multiple models. This paper considers to select the best hypothesis from multiple candidates produced by a single model, which is often called reranking.
>
> In footnote 2 we clarify the use of the term "ensembling". We chose to use "ensembling" instead of "reranking" since we also experiment with non-reranking techniques such as GenerateBest. While the best results are obtained with reranking, we believe that the inclusion of GenerateBest (and the finding that it is not competitive) is an important contribution of the paper.

---

### Official Review · Reviewer_aEq9 · 2023-08-04

**Soundness:** 4

**Excitement:**

4: Strong: This paper deepens the understanding of some phenomenon or lowers the barriers to an existing research direction.

**Paper Topic And Main Contributions:**

The paper studies various techniques for ensembling translations from large language models. Experiments were conducted on several LLM models (ChatGPT, LLaMA, Alpaca) with 8 directions from English to four languages (DE,RU,CS,UK) and vice versa. The results show that ensembling increases translation quality and reduces hallucination. It is closing the gap between the studied models and ChatGPT. The results also show that generating a final translation with MBR decoding achieves high scores.

**Reasons To Accept:**

The empirical study is comprehensive and covered many areas.

The paper is well written and easy to follow.

**Reasons To Reject:**

I do not have a reason.

**Reproducibility:**

4: Could mostly reproduce the results, but there may be some variation because of sample variance or minor variations in their interpretation of the protocol or method.

**Reviewer Confidence:**

3: Pretty sure, but there's a chance I missed something. Although I have a good feel for this area in general, I did not carefully check the paper's details, e.g., the math, experimental design, or novelty.

---

> ### Author Rebuttal · Authors · 2023-08-28
>
> Thank you for your positive reviews! We are glad that you found the empirical study comprehensive and the paper well written and easy to follow.

---

### Official Review · Reviewer_uVRh · 2023-08-05

**Soundness:** 4

**Excitement:**

4: Strong: This paper deepens the understanding of some phenomenon or lowers the barriers to an existing research direction.

**Paper Topic And Main Contributions:**

In this paper, the authors studied the topic of improving the quality of the generated text for the specific problems of Large Language Model (LLM)-based machine translation by hypothesis embeddings, such as hallucination and unreliable output. Several comparable results were conducted through experimental pairs such as open-source scenario vs. closed-source one, different languages transformation, and searching strategies.

The main contribution of this paper is that, through a set of experiments, three findings were given: translation quality can be enhanced with a small number of samples; instruction tuning affects the relationship between the diversity of the hypotheses and the sampling temperature; hypothesis ensembling reduces the number of generated hallucinations.

**Questions For The Authors:**

What is the translation performance of closed-source LLM like ChatGPT?
If the hypotheses of powerful LLMs ensemble with results of LLMs with relatively small scale parameters, for example, LLaMA2-70B ensembles with Alpaca and LLama1-7B, the ensembling translation quality will drop?

**Reasons To Accept:**

The main strengths of this paper are:
1. the authors demonstrated clear structure and logic, which is easy to follow;
2. adequate comparisons among each set of experiments were explained in detail with tables and graphs;
3. the problem the authors focused on is an important and state-of-the-art method for machine translation using the LLM paradigm.

**Reasons To Reject:**

The weakness of this paper is that I found their study is limited to translation pairs among languages in European, thus the generalization between other language pairs, for example, the ‘European-Asian’ pair, is unknown.

**Reproducibility:**

5: Could easily reproduce the results.

**Reviewer Confidence:**

5: Positive that my evaluation is correct. I read the paper very carefully and I am very familiar with related work.

---

> ### Author Rebuttal · Authors · 2023-08-28
>
> Thank you for your positive reviews! We will now try to address your concerns about our paper.
>
> >Their study is limited to translation pairs among languages in European, thus the generalization between other language pairs, for example, the ‘European-Asian’ pair, is unknown.”
>
> We followed your suggestion and  ran some experiments on translation from English to Chinese on the WMT 2022 test set. The results are inline with our findings for non-Asian languages. The results in Table 1 can be complemented with the following information for En-Zh:
>
> | Model   | Method   | N  | COMET     | BLEURT    |
> |---------|----------|----|-----------|-----------|
> | ChatGPT | Sampling | 1  | 86.27     | 71.34     |
> | ChatGPT | Greedy   | 1  | 86.94     | 72.22     |
> | ChatGPT | Ranking  | 20 | 87.86     | 72.69     |
> | ChatGPT | MBR      | 20 | 88.28 | 72.96 |
> | ChatGPT | Ranking  | 50 | 88.08     | 72.78     |
> | ChatGPT | MBR      | 50 | **88.47** | **72.98** |
>
> > What is the translation performance of closed-source LLM like ChatGPT?
>
> The results for ChatGPT are included in Table 1 (see the first two lines for the performance of single hypothesis translations). While ChatGPT’s performance alone is good (as reported by, e.g., Hendy et al. (2023)), we show that it can be further boosted by ensembling multiple hypotheses, especially for En-X translation (see Figure 1).
>
> > If the hypotheses of powerful LLMs ensemble with results of LLMs with relatively small scale parameters, for example, LLaMA2-70B ensembles with Alpaca and LLama1-7B, the ensembling translation quality will drop?
>
> This is a very interesting question. We ran some experiments and found that combining the outputs of a powerful LLM (e.g., ChatGPT) and the outputs of a smaller scale one (e.g., Alpaca 7B) with MBR decoding actually *improves* the final quality of the ensemble. For some language pairs (En-De and En-Ru) a combination of 20 samples from each model is even better than an ensemble of 50 samples from the stronger model (see some results below). We believe this might have to do with the higher diversity of the hypotheses. We will add this new experiment to the final version of the paper.
>
> | Model               | Method | N         | En-De | En-Ru | En-Cs | En-Uk |
> |---------------------|--------|-----------|-------|-------|-------|-------|
> | ChatGPT             | Greedy | 1         | 87.01 | 87.77 | 90.04 | 87.66 |
> | ChatGPT             | MBR    | 20        | 88.09 | 89.41 | 91.97 | 90.03 |
> | ChatGPT + Alpaca 7B | MBR    | 20+20=40  | 88.49 | 89.89 | 92.15 | 90.23 |
> | ChatGPT             | MBR    | 50        | 88.25 | 89.64 | 92.21 | 90.31 |
> | ChatGPT + Alpaca 7B | MBR    | 50+50=100 | **88.62** | **90.06** | **92.38** | **90.58** |

---

### Meta-Review · Area_Chair_SR3N · 2023-09-05

**Recommendation:** 4
**Best Paper Recommendation:** No

**Metareview:**

Paper on combination of MT outputs generated by LLMs. Different strategies to generate multiple hypotheses with LLMs, as well as methods to produce the final translation are explored. Contributions of the paper are: ensembling multiple hypotheses significantly improves translation quality;   instruction tuning affects diversity/temperature relationship; hypothesis ensembling reduces hallucinations. Experiments are provided for 8 directions of 4LPs. Paper is well written and well-organized paper.


Reasons to accept:
Paper investigates MT with LLMs in a scientifically solid way: they use both closed and open source models; the former to see performance impact on high end models, the latter to allow for reproducibility of results and more in depth investigations.
Experimental results are informative and performed with up to date evaluation metrics.

Reasons to reject:
Limited language coverage (4 language pairs)
Limited novelty in the approaches, the authors apply (and properly cite) known methods for their experiments.
Some methods are just tried and do not work consistently across models and metrics making this work sound very empirical. We do not know if these results will still hold for another version of the same models.
The impact of instruction tuning is measured on one single model and LP.

Detailed comments
Figure 1: the type of plot is misleading as it suggests a continuum between languages which does not exist.
269: “significantly increases  the cost”. Cost of ChatGPT should not be part of the equation. If you are using external LLMs you should assume they are for free.
303 Understanding the prompt requires reading the appendix.

**Meta-Review:**

Overall, this is looks to be a solid paper which sets some good practice when handling LLMs.  Although results look shaky, there are some clear patterns emerging, which are put in evidence by the authors. Definitely better than papers which purely analyze behavior of proprietary inaccessible LLMs.

---

### Decision · Program_Chairs · 2023-10-07

**Decision:**

Accept-Main

**Comment:**

Paper on combination of MT outputs generated by LLMs. Different strategies to generate multiple hypotheses with LLMs, as well as methods to produce the final translation are explored. Contributions of the paper are: ensembling multiple hypotheses significantly improves translation quality;   instruction tuning affects diversity/temperature relationship; hypothesis ensembling reduces hallucinations. Experiments are provided for 8 directions of 4LPs. Paper is well written and well-organized paper.


Reasons to accept:
Paper investigates MT with LLMs in a scientifically solid way: they use both closed and open source models; the former to see performance impact on high end models, the latter to allow for reproducibility of results and more in depth investigations.
Experimental results are informative and performed with up to date evaluation metrics.

Reasons to reject:
Limited language coverage (4 language pairs)
Limited novelty in the approaches, the authors apply (and properly cite) known methods for their experiments.
Some methods are just tried and do not work consistently across models and metrics making this work sound very empirical. We do not know if these results will still hold for another version of the same models.
The impact of instruction tuning is measured on one single model and LP.

Detailed comments
Figure 1: the type of plot is misleading as it suggests a continuum between languages which does not exist.
269: “significantly increases  the cost”. Cost of ChatGPT should not be part of the equation. If you are using external LLMs you should assume they are for free.
303 Understanding the prompt requires reading the appendix.